# Prepectoral vs. Submuscular Immediate Breast Reconstruction in Patients Undergoing Mastectomy after Neoadjuvant Chemotherapy: Our Early Experience

**DOI:** 10.3390/jpm12091533

**Published:** 2022-09-19

**Authors:** Lorenzo Scardina, Alba Di Leone, Ersilia Biondi, Beatrice Carnassale, Alejandro Martin Sanchez, Sabatino D’Archi, Antonio Franco, Francesca Moschella, Stefano Magno, Daniela Terribile, Damiano Gentile, Alessandra Fabi, Anna D’Angelo, Liliana Barone Adesi, Giuseppe Visconti, Marzia Salgarello, Riccardo Masetti, Gianluca Franceschini

**Affiliations:** 1Breast Unit, Department of Women, Children and Public Health Sciences, Fondazione Policlinico Universitario Agostino Gemelli IRCCS, 00168 Rome, Italy; 2Multidisciplinary Breast Center, Fondazione Policlinico Universitario Agostino Gemelli IRCCS, Largo Agostino Gemelli 8, 00168 Rome, Italy; 3Breast Unit, IRCCS Humanitas Research Hospital, 20089 Milan, Italy; 4Precision Medicine Breast Unit, Scientific Directorate, Department of Women, Children and Public Health Sciences, Fondazione Policlinico Universitario Agostino Gemelli IRCCS, 00168 Rome, Italy; 5Department of Diagnostic Imaging, Oncological Radiotherapy and Hematology, Division of Breast Surgery, Fondazione Policlinico Universitario Agostino Gemelli IRCCS, 00168 Rome, Italy

**Keywords:** breast cancer, mastectomy, neoadjuvant chemotherapy, immediate prosthetic breast reconstruction

## Abstract

Background: Conservative mastectomy with immediate prosthetic breast reconstruction (IPBR) is an oncologically accepted technique that offers improved esthetic results and patient quality of life. Traditionally, implants have been placed in a submuscular (SM) plane beneath the pectoralis major muscle (PMM). Recently, prepectoral (PP) placement of the prosthesis has been increasingly used in order to avoid morbidities related to manipulation of the PMM. The aim of this study was to compare outcomes of SM vs. PP IPBR after conservative mastectomy in patients with histologically proven breast cancer treated with neoadjuvant chemotherapy (NAC). Methods: In this retrospective observational study, we analyzed two cohorts of patients that underwent mastectomy with IPBR after NAC in our institution from January 2018 to December 2021. Conservative mastectomy was performed in 146 of the 400 patients that underwent NAC during the study period. Patients were divided into two groups based on the positioning of implants: 56 SM versus 90 PP. Results: The two cohorts were similar for age (mean age 42 and 44 years in the SM and PP group respectively) and follow-up (33 and 20 months, respectively). Mean operative time was 56 min shorter in the PP group (300 and 244 min in the SM and PP group). No significant differences were observed in overall major complication rates. Implant loss was observed in 1.78% of patients (1/56) in the SM group and 1.11% of patients (1/90) in PP group. No differences were observed between the two groups in local or regional recurrence. Conclusions: Our preliminary experience, which represents one of the largest series of patients undergoing PP-IPBR after NAC at a single institution documented in the literature, seems to confirm that PP-IPBR after NAC is a safe, reliable and effective alternative to traditional SM-IPBR with excellent esthetic and oncological outcomes; it is easy to perform, reduces operative time and minimizes complications related to manipulation of PPM. However, this promising results need to be confirmed in prospective trials with longer follow-up.

## 1. Introduction

Conservative mastectomy with immediate prosthetic breast reconstruction (IPBR) for selected patients is considered an oncologically safe technique that permits enhanced quality of life and esthetic outcomes [1].

Traditionally, the prosthesis has been positioned under the pectoralis major muscle (PMM) in a submuscular (SM) pocket. Recently, a new technique has been used in which implants have been placed in the prepectoral (PP) plane [2,3].

This innovative procedure reduces complications related to creation of the SM pocket and improves esthetic outcomes [4,5].

Patients affected by lymph node-positive disease, high tumor-to-breast volume ratio, and aggressive biological features (high grade, triple negative, HER2-positive) are frequently candidates for neoadjuvant chemotherapy (NAC) [6,7].

All NAC agents are cytotoxic and can kill proliferative cells and cause adverse effects on the immune system, so they may theoretically affect surgical outcomes (fat necrosis, wound dehiscence, hematoma and infection) [8].

The aim of this study was to analyze 146 consecutive conservative mastectomy and IPBR operations using a prepectoral approach from January 2018 to December 2021 in patients with breast cancer treated with NAC in our institution.

## 2. Materials and Methods

All consecutive breast cancer patients undergoing conservative mastectomy followed by IPBR after NAC between January 2018 and December 2021 were reviewed.

A total of 146 patients were treated by unilateral or bilateral conservative mastectomy over the study period. We divided our sample into 2 cohorts based on the site of implant placement: 56 SM versus 90 PP.

Data were recorded in order to evaluate operative time, postoperative major complications (implant explantation), oncological outcomes (local recurrence, disease free survival and overall survival).

Results are expressed as means with associated median and range. Statistical significance was set at *p* < 0.050. The Fisher exact test was used for comparison of categorical variables. Data analyses were performed with IBM SPSS 24.0 software.

### Methodological Approach

NAC is used in patients with contraindications for surgery, locally advanced breast cancer and inflammatory breast cancer to downstage large tumors, improve local and distant site disease control and to increase breast conserving approaches [9,10].

Therapeutic regimens of NAC include anthracyclines, cyclophosphamide and taxanes or carboplatin; taxanes are combined with targeted trastuzumab therapy in case of HER2-positivity.

A multidisciplinary meeting was held with a meticulous staging of tumor and careful selection of patients with clinical assessment, ultrasonography, mammography and RM. Specific evidence-based guidelines were released to ensure that each patient treated in the neoadjuvant setting may receive the most effective, evidence-based chemotherapy regimen, in a personalized, multidisciplinary background.

All patients receiving an indication to undergo NAC were taken care of by a “neoadjuvant oncologic treatment team”, which explained the care plan designed by the multidisciplinary panel, and a “neoadjuvant supportive care team”, which directed patients for a complete psychological, nutritional and lifestyle evaluation. Therefore, every specific treatment was tailored to each patient in a multidisciplinary holistic fashion [11,12].

Conservative mastectomy was evaluated for patients for whom, although undergoing NAC, breast conserving surgery (BCS) could not ensure oncological control and good esthetic outcomes [13].

Indications included impossibility to obtain oncologically safe margins with BCS and large tumor dimension respect to breast size. The following exclusion criteria to conservative mastectomy were used: inflammatory and locally advanced breast cancer, obesity (BMI > 30 kg/m^2^) previous radiotherapy, active smoking.

Before surgery each patient underwent an examination by a 3-stage breast tissue coverage classification. A digital mammogram allowed an accurate evaluation of the breast coverage and a preview of the resulting flap thickness, with a consequent possible prevision of flap quality and vascularization [14].

After mastectomy, the type of reconstruction (SM vs. PP) was based on flap thickness and perfusion, which were assessed with indocyanine green dye fluoroangiography and photo dynamic eye (PDE) imaging system [15].

Operative time was recorded.

Major complications (implant removal), loco-regional recurrences and esthetic outcomes were assessed in all patients [16].

An Automated Breast Volume Scanner (ABVS), a computer-based system for the appraisal of whole breast, allowed us to acquire 3D ultrasound images that can allow examination in multiplanar reconstructions thicker skin flaps of patients with PP-IPBR [17].

## 3. Results

In the 3 year study period from January 2018 to December 2021, 146 consecutive patients underwent conservative mastectomy after NAC with IPBR.

Patient characteristics were similar in the SM and PP groups (Table 1). Mean age was 42 (28–69) and 44 (28–66) years respectively.

SM-IPBR was carried out in 56 patients and PP-IPBR in 90 patients.

Nipple-sparing mastectomy was carried out on 135 patients (70 unilateral and 64 bilateral); on 8 patients skin-sparing mastectomy was performed (5 unilateral and 3 bilateral); only 3 patients received a bilateral skin-reducing mastectomy.

Therapeutic unilateral conservative mastectomy was performed on 75 patients; bilateral mastectomy with IPBR was performed on 71 patients.

Simultaneous contralateral symmetrization was performed in 18/22 (81.82%) patients of the SM group and in 15/53 (28.31%) patients of the PP group. Surgical treatment is summarized in Table 2.

Mean operative time of bilateral mastectomy with PP-IPBR was 267 min and 314 min for SM-IPBR, while it was 228 min and 281 min for unilateral, respectively.

The median follow-up was similar: 33 months (2–48) in the SM group and 20 months (1–48) in the PP group. No significant difference was observed in overall complication rates and oncological outcomes between the two reconstructive cohorts.

Only one patient per group (respectively 1.11% PP and 1.80% SM) lost an implant due to infection (Table 3). These complications can be classified as Clavien-Dindo grade III.

During follow-up, nipple–areola complex recurrence occurred only in 1/56 (1.78%) patients of the SM group. Local recurrences occurred in 2/56 (3.57%) patients in the SM group and in 1/90 patients (1.11%) of the PP cohort.

As concerns disease-free survival, 1/56 (1.78%) of patients of the SM group and 1/90 (1.11%) patients of the PP group with triple negative breast cancer both developed brain metastases 6 and 12 months, respectively, after surgery.

## 4. Discussion

This investigation constitutes one of the largest patient series of a single institution studies in the literature on the application of this novel technique.

During NAC, clinical complications and psychological issues can occur, undermining outcomes. In our breast unit, we conduct a multidisciplinary method to simplify the application of evidence-based oncologic protocols and improve patient quality of life and we offer IPBR to all patients undergoing conservative mastectomy.

NAC is generally used in patients with locally advanced breast cancer and inflammatory breast cancer to downstage large tumors and to improve local and distant site disease control, which has led to increases in the breast conserving approach [18,19]. Patients with T1, N0 or M0 triple-negative or HER2-positive breast cancer often undergo NAC because these cancers are usually sensitive to chemotherapy [20].

However, a large number of patients receiving NAC undergo mastectomy, either because breast conserving surgery is not feasible or because of patient preference [19].

NAC can complicate surgical wound care following mastectomy and IPBR.

One of the most frequent postoperative morbidities following breast cancer surgery is wound infection [21] and a common side effect of NAC is neutropenia [22], so these patients can be subjected to an increased risk of postoperative complications.

Kasushik et al. [23] investigated wound complications and concluded that the rate of complications was similar in groups with and without NAC.

The safety of reconstruction after NAC has improved in recent years [10]. Varghese et al. found that there was no significant difference in rates of complications or delay to adjuvant therapy among women with and without NAC [24]. Other studies have reported that the complication rate of IPBR after NAC is the same as that without NAC [25,26,27,28]. In our series comparing the two cohorts there was no significant difference in overall major complication rates.

Conservative mastectomy (skin-sparing or nipple-sparing) with PP-IPBR is an oncologically accepted technique that allows improved cosmetic outcomes and patient quality of life [29,30,31] (Figure 1).

Generally implants have been positioned into the SM plane behind the coverage of PMM to minimize surgical complications, mitigate implant visibility, palpability and ripple [32]. For many years, we have used only SM placement of the implants but since 2016, we have started to perform PP-IPBR using a Polytech implant with a micropolyurethane foam-coated shell surface (microthane) that does not require further ADM coverage.

Traditionally, unilateral conservative mastectomy allows avoiding contralateral breast symmetrization: in our experience, a symmetrization procedure was performed in 17/21 (80.0%) patients in the SM group, compared to only 14/52 (26.0%) cases in the PP group, so there was observed a significant difference in these terms (*p* = 0.03).

Surgical expertise and careful patient selection are the basis to achieve an excellent result with PP-IPBR [33,34]. During surgery, thickness and perfusion of skin flaps should be evaluated, in addition to individual risk factors [35], before choosing the type of reconstruction (PP vs. SM) [36,37].

Similar results in terms of surgical complications and oncological outcomes should be obtained with either the SM or PP technique [38]. In our series, we found no statistically significant differences in terms of implant failure or recurrence between the two cohorts.

In our latest work, we showed how PP-IPBR is a safe, effective and cost-effective alternative to the SM technique with similar complications rates, cosmetic results and quality of life [5].

Postoperative complications observed in our study were in line with the literature: only two implant failures were recorded.

To date we assume that, when technically feasible, in selected cases, PP-IPBR after NAC should be considered an excellent surgical option.

As regards oncological safety, one local relapse occurred in the nipple in the SM group; two local recurrences were observed in the SM group and one in the PP group during follow-up.

Disease-free survival was similar: one patient in the SM group 1/56 (1.78%) and one patient in the PP group 1/90 (1.11%) developed brain metastases 6 and 12 months, respectively, after surgery.

Follow-up for these patients was carried out in our unit by conventional breast ultrasound and digital mammography, but also with 3D-ABVS, which permitted a careful examination of skin flaps.

In our previous study, we described encouraging and satisfactory results with PP-IPBR in terms of cosmetic results, chronic pain, shoulder dysfunction, sports activity, sexual and relationship aspects and skin sensibility [39,40].

This study constitutes the largest series of patients undergoing PP-IPBR after NAC; however, it is necessary to underline that it has some limitations due to retrospective analysis, short follow-up time and the missed evaluation of minor complications.

## 5. Conclusions

All the data published in our previous studies have been confirmed by these new results and provide additional evidence in support of PP-IPBR for high-risk and more fragile patients.

Our model can encourage clinicians to personalize supportive care and direct it towards precision medicine. The development of an appropriate clinical pathway, with multidisciplinary competence and the performance of standardized tasks, is mandatory to obtain a successful treatment in the neoadjuvant setting.

In conclusion, our experience indicates that PP-IPBR can represent a valid alternative to traditional SM-IPBR, improving outcomes and patient quality of life.

Careful patient selection, adequate surgical experience and appropriate surgical training are essential to improve outcomes and reduce complications.

Further prospective trials with a larger number of patients and longer follow-up are necessary to draw more validated conclusions.

## Figures and Tables

**Figure 1 jpm-12-01533-f001:**
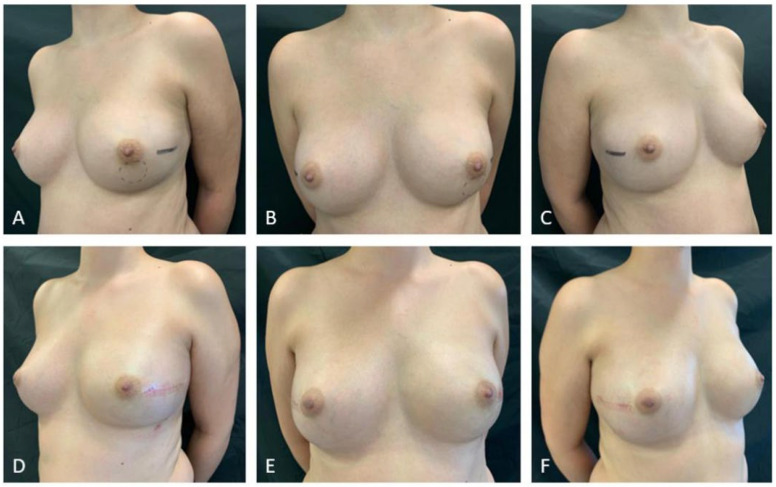
Pictures of a 45-year-old bilateral breast cancer patient. (**A–C**) Preoperative pictures. (**D–F**) Three-month postoperative pictures after bilateral nipple-sparing mastectomy and direct-to-implant prepectoral reconstruction after NAC.

**Table 1 jpm-12-01533-t001:** Patient characteristics and clinicopathological features.

	Total	PP-IPBR	SM-IPBR	P
**Patient N**	146	90 (61.65%)	56 (38.35%)	
Age (years)	43 (27–69)	44 (28–66)	42 (28–69)	
**Histotype**				0.15
- Invasive ductal	136 (93.15%)	83 (92.22%)	53 (94.65%)	0.57
- Invasive lobular	10 (6.85%)	7 (7.78%)	3 (5.35%)	
**Grading**				
- 1	0	0	0	
- 2	30 (20.55%)	16 (17.78%)	14 (25.00%)	0.29
- 3	116 (79.45%)	74 (82.22%)	42 (75.00%)	
Ki67 (%)	46 (1–96)	45 (1–96)	46 (5–80)	0.86
**Stage**				
- pT1	25 (17.12%)	16 (17.77%)	9 (16.08%)	0.13
- pT2	94 (64.38%)	53 (58.88%)	41 (73.22%)	
- pT3	22 (15.07%)	16 (17.77%)	6 (10.72%)	
- pT4	5 (3.42%)	5 (5.55%)	0	
- pN0	56 (38.35%)	33 (36.66%)	23 (41.08%)	0.21
- pN1	64 (43.83%)	37 (41.11%)	27 (48.22%)	
- pN2	23 (15.75%)	17 (18.88%)	6 (10.72%)	
- pN3	3 (2.05%)	3 (3.33%)	0	
- Multifocal	60 (41.09%)	33 (36.66%)	27 (48.22%)	0.17
- pCR	28 (19.18%)	15 (16.67%)	13 (23.21%)	0.33
**Biological subtypes**				
- Luminal-like	61 (41.78%)	37 (41.12%)	24 (42.86%)	**0.059**
- HER2-enriched	51 (34.94%)	37 (41.12%)	14 (25.00%)	
- Triple negative	34 (23.29%)	16 (17.77%)	18 (32.15%)	
**BRCA 1/2**	34 (23.29%)	20 (22.22%)	14 (25.00%)	0.70
**Postoperative treatment**				
- Radiotherapy	48 (32.87%)	28 (31.12%)	20 (35.72%)	0.20
- Hormone therapy	54 (36.99%)	31 (34.45%)	23 (41.08%)	
- Chemotherapy	8 (5.48%)	2 (2.23%)	6 (10.72%)	

**Table 2 jpm-12-01533-t002:** Detailed surgical treatment.

	Total	PP-IPBR	SM-IPBR	P
**Mastectomy**				
- Monolateral	75 (51.37%)	53 (58.88%)	22 (39.29%)	**0.021**
- Bilateral	71 (48.63%)	37 (41.12%)	34 (60.71%)	
**Surgical procedures**				
- Nipple Sparing	135 (92.47%)	81 (90.00%)	54 (96.43%)	0.15
- Skin Sparing	8 (5.48%)	6 (6.67%)	2 (3.57%)	
- Skin Reducing	3 (2.05%)	3 (3.33%)	0	
**Contralateral symmetrization**	33 (44.00%)	15 (28.31%)	18 (81.82%)	**0.030**

**Table 3 jpm-12-01533-t003:** Complications.

	Total	PP-IPBR	SM-IPBR
**Major complications**			
Implant explanation	2 (1.37%)	1 (1.11%)	1 (1.80%)

## Data Availability

Data could be found asking to the corrisponding author (email).

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
