# Peer review of "Prepectoral vs. Submuscular Immediate Breast Reconstruction in Patients Undergoing Mastectomy after Neoadjuvant Chemotherapy: Our Early Experience"

_jpm, 2022, doi:10.3390/jpm12091533_

Round 1
Reviewer 1 Report
The manuscript deals with an interesting topic. It is important to provide evidence that the position of the implant in breast reconstruction is less important than the patient might think. I congratulate the authors on their good results. 1.11% PP and 1.80% SM implant loss due to infection is very low. I have 2 suggestions to improve the work: Please describe in a table which complications occurred postoperatively and how often they occurred. Please classify the postoperative complications according to the Clavien- Dino classification.
Author Response
Dear Editors,
We are pleased to receive your corrections that could make our paper entitled “Prepectoral vs submuscular immediate breast reconstruction in patients undergoing mastectomy after neoadjuvant chemotherapy: our early experience” more suitable for publication in “Journal of Personalized Medicine”.
- In the following text, you will find our comments for each reviewer. (Our answers are reported according to the precise order of the questions raised by each reviewer)
- Reviewer #1:
Thank you for the suggestions of the Reviewer #1 to improve the quality of our paper. We appreciated your important contribution.
Table of major complications added at the end with Calvien-Dindo classification grade in the text.
Reviewer 2 Report
In this manuscript, the authors compared the outcomes of immediate breast reconstruction between the placement of the implants in the submuscular vs prepectoral plane.
The manuscript is well-written, the methodology is adequate, and the results are promising, and they suggested the possibility to perform further studies. However, there are some comments that should be made to improve the scientific quality of the manuscript:
- In the material and methods section, the type of statistical analysis that has been performed for each of the variables should be added (chi square, etc.) (p values in Table 1).
- The authors properly mention that their article has some limitations (retrospective, short follow-up). However, there are some more limitations since they have not compared other complications between the groups that are mentioned in the discussion (haematomas, infections, skin necrosis, rippling, capsular contracture, etc.) or patient satisfaction.
- There were statistical analysis for assessing if both groups are comparable, which they are. However, there are not statistical analyzes for the results (differences in surgical time, loss implant, disease free survival), which greatly reduces the power of the study and does not allow confirming whether there are differences between the two groups. If possible, I suggest to add these statistical analyzes.
Finally, I consider this is an interesting study and well performed, so with some corrections I consider that it can be appropriate for publication in this journal.
Author Response
Dear Editors,
We are pleased to receive your corrections that could make our paper entitled “Prepectoral vs submuscular immediate breast reconstruction in patients undergoing mastectomy after neoadjuvant chemotherapy: our early experience” more suitable for publication in “Journal of Personalized Medicine”.
- In the following text, you will find our comments for each reviewer. (Our answers are reported according to the precise order of the questions raised by each reviewer)
- Reviewer #2:
We appreciated the punctual observations of the Reviewer #2.
1) Type of statistical analysis added.
2) Limitations added.
3) In the two cohorts regarding the above variables (differences in surgical time, loss implant, disease free survival), no sufficient events were documented in order to proceed with the statistical analysis (example 1/56 vs 1/90). It is therefore necessary to collect more patients and to obtain an adequate follow-up.
Reviewer 3 Report
We cheer the authors for this interesting study that may open new perspectives of investigation and discussion about IPBR. I would like to give the authors few suggestions before publishing the article.
1) Introduction
1.1) Page 2, line 55-58: The authors state quite clearly the goal of their study, nevertheless is hard to understand here how NAC should affect the results of a surgical procedure such as IPBR. Could they explain this briefly?
2) Discussion:
2.1) Page 4, line 149: The authors state “NAC can complicate surgical wound following mastectomy and IPBR”. Could they explain the patophysiology of this sentence? How NAC may increase complications?
2.2) Page 4, line 158: As the authors state: “Conservative mastectomy (skin sparing or nipple sparing) with PP - IPBR is an oncologically accepted technique that allows to improve cosmetic outcomes and patient quality of life” they may cite a more recent paper that investigates the oncologial safety of the PP breast reconstruction: “Casella D, Kaciulyte J, Resca L et al. Looking beyond the prepectoral breast reconstruction experience: a systematic literature review on associated oncological safety and cancer recurrence incidence. European Journal of Plastic Surgery volume 45, pages223–231 (2022)”.
2.3) Page 5, line 168: The authors do not explain how they perfom PP-IPBR nor here nor in the Methods section. They should specify if they use some peri-prosthetic devices like ADM or meshes as some post-operative complications’ incidence may be linked to that.
2.4) Page 5, line 175: “During surgery thickness and perfusion of skin flaps should 174 be evaluate before choosing the type of reconstruction (PP vs SM)”. As the authors correctly state, skin flap viability is of course a fundamental criteria, but other factors may be taken into consideration when choosing between PP and SM (Casella D, Kaciulyte J, Lo Torto F, et al. "To Pre or Not to Pre": Introduction of a Prepectoral Breast Reconstruction Assessment Score to Help Surgeons Solving the Decision-Making Dilemma. Retrospective Results of a Multicenter Experience. Plast Reconstr Surg. 2021 Jun 1;147(6):1278-1286.).
Author Response
Dear Editors,
We are pleased to receive your corrections that could make our paper entitled “Prepectoral vs submuscular immediate breast reconstruction in patients undergoing mastectomy after neoadjuvant chemotherapy: our early experience” more suitable for publication in “Journal of Personalized Medicine”.
- In the following text, you will find our comments for each reviewer. (Our answers are reported according to the precise order of the questions raised by each reviewer)
Reviewer #3:
Thank you for the suggestions of the Reviewer #1 to improve the quality of our paper.
1) Explained in the introduction.
2) Explained the the patophysiology of that sentence.
3) Reference cited
4) Specified the surgical technique
5) Others risk factors added with reference